# iCan, Empowering Recovery: Evaluating a Patient-Centred Cancer Rehabilitation Programme across the Cancer Care Continuum

**DOI:** 10.3390/diseases12100236

**Published:** 2024-10-02

**Authors:** Thomas A. Loweth, Suzan R. Taylor, Gareth Mapp, Kim Bebbington, Naomi Atkin, Chris Kite

**Affiliations:** 1Faculty of Health, Medicine and Society, Division of Public Health, Sport and Wellbeing University of Chester, Chester CH1 4BJ, UK; tl251@leicester.ac.uk (T.A.L.); suzan.taylor@chester.ac.uk (S.R.T.); 2Diabetes Research Centre, College of Life Sciences, University of Leicester, Leicester LE5 4PW, UK; 3Lifestyle Fitness & Physiotherapy, Castle Court, Whittington, Shrewsbury SY11 4DF, UK; gareth.mapp@gmail.com (G.M.); kimbebbington.lifestyle@gmail.com (K.B.); 4Lingen Davies Cancer Fund Charity, Hamar Centre, Royal Shrewsbury Hospital, Shrewsbury SY3 8XQ, UK; naomi.atkin@lingendavies.co.uk; 5School of Health and Society, Faculty of Education, Health and Wellbeing, University of Wolverhampton, Wolverhampton WV1 1LY, UK; 6Warwickshire Institute for the Study of Diabetes, Endocrinology and Metabolism (WISDEM), University Hospitals Coventry and Warwickshire NHS Trust, Coventry CV2 2DX, UK

**Keywords:** cancer rehabilitation, holistic care, mindfulness, functional capacity, physical activity, mental wellbeing, evaluation

## Abstract

Background/Objectives: The adverse effects of cancer and the long-term sequelae of associated treatments result in reduced quality of life and increased mortality for patients. Supporting patients with cancer to mitigate adverse outcomes is an important aspect of oncology care and the primary purpose of cancer rehabilitation. A retrospective service evaluation was conducted to evaluate the effectiveness of the core iCan patient-centred cancer rehabilitation service. Methods: At the beginning and end of a five-week programme, a series of questionnaires evaluating changes in mental health and wellbeing, and physical activity performance/attitudes, and functional capacity were administered to participants. Results: Following iCan, we found that functional capacity was improved (30 s sit-to-stand: +6.3 repetition; *d* = −1.00, *p* < 0.001) and that self-reported physical activity was increased (~1173 MET-mins/wk; *d* = −0.76, *p* < 0.001); participants also perceived greater capability, opportunity, and motivation to be active. Mental wellbeing was also improved (SWEMWBS: *d* = −0.69, *p* < 0.001), whilst fatigue was reduced (FACIT: *d* = −0.77, *p* < 0.001). Conclusion: It appears that iCan has beneficial effects upon the physical/functional and psychological health of its participants. Where data are available, there appear to be clinically significant improvements across the range of measured functional, wellbeing, and activity/sedentariness outcomes, which suggest that participation in iCan is instrumental in adding value to the health and wellbeing of patients.

## 1. Introduction

Cancer is a chronic, life-limiting disease which is typically characterized by pain and a range of complex symptoms such as heightened fatigue, reduced strength, dyspnea, nausea, and mental health disorders, which negatively impact both quality and length of life [1]. In the UK, over 350,000 new cancer cases are diagnosed each year and given the projected increase in cancer incidence rates, this figure is expected to reach ~514,000 by 2035 [2,3]. Approximately 4570 people in Shropshire, Telford and Wrekin, and Powys are diagnosed with cancer each year, with an estimated additional 50,000 people currently living with and beyond cancer in the region and further increases predicted [4,5].

The Lingen Davies Cancer Fund Charity is dedicated to making a positive difference to the lives of people impacted by cancer in Shropshire, Telford and Wrekin, and Mid-Wales by enhancing cancer services and improving lives in the community. The strategies in place and developing across the localities for achieving these outcomes include ensuring every person affected by cancer is offered high-quality information, support, advice, and personalised care specific to their cancer and their needs. Actions are set out to promote, encourage, and empower people to adopt healthier lifestyles to reduce risks and improve long-term outcomes for those diagnosed with cancer and to provide individualized care and support to cancer patients along all parts of the pathway.

The adverse effects of cancer and the long-term sequelae of anti-cancer treatments (i.e., chemotherapy, radiotherapy, and surgery) often result in patients with cancer suffering from several repercussions (e.g., weight loss, cancer cachexia, declines in functional capacity, and reduced mental wellbeing) that can combine to reduce quality of life (QoL) [6,7]. Furthermore, despite cancer-related mortality rates having declined by 24% over the past 40 years, the 10-year survival rate of cancer patients is 50%, and the disease still contributes towards 167,000 deaths per annum in the UK [2]. As such, implementing approaches that further reduce mortality rates, mitigate these adverse outcomes, and support cancer patients during treatment is an important aspect of oncology care and the primary purpose of cancer rehabilitation services [8,9].

Rehabilitation is defined as “a set of interventions designed to optimize functioning and reduce disability in individuals with health conditions, in interaction with their environment” [10]. More specifically, the aim of cancer rehabilitation is to help patients regain functioning and social participation [11]. Cancer rehabilitation is recommended in numerous clinical guidelines and can comprise diverse interventions such as physical therapy and activity (e.g., exercise, yoga, lymphatic drainage, etc.), supportive medications (e.g., for pain or insomnia), psychological interventions (e.g., resiliency training, coping strategies, relaxation techniques, etc.), and assistance for social (re)integration (e.g., preparing to return to work) [12]. Optimal cancer care is often considered to take place across a continuum (i.e., early detection, diagnosis, treatment, survivorship, palliative care, and end of life) [13], and rehabilitation is an integral component of that continuum; it should begin at diagnosis and continue during and beyond treatment [14].

Cancer rehabilitation services (including exercise prescription) have been shown to enhance the quality of cancer survivorship in various cancer populations by providing targeted interventions that encourage participants to socialize with others whilst taking part in activities that promote physical and psychological health and improve functional capacity and health-related QoL [15]. A positive effect has been shown for cancer patients at all stages of disease, reducing fatigue, improving tolerance to treatment, significantly improving recovery time and sleep, and enhancing cardiovascular and immune function and health-related QoL [16].

A major strength of cancer rehabilitation is that it may address symptoms and impairments in a holistic manner—considering the person’s goals, strengths, and contextual factors—and address function and participation in everyday life. Rehabilitation interventions are contextual, and necessarily so, to achieve individualization in treatment [17]. It is also noted that, whilst the prevalence of cancer and associated morbidities requires the development of integrated models, the needs of patients may not be met by current cancer rehabilitation services [16]. These vary across the world; there is little standardization of programme content, there are specific tumour site limitations and a wide range of challenges/barriers to successful implementation (e.g., staffing, structure, referrals, and delivery setting).

A key component of cancer rehabilitation comes from supporting participants to be more physically active [7]. Indeed, it is well documented that engaging in regular physical activity (PA) and exercise promotes a wide range of health-related benefits for cancer patients (e.g., improved cardiovascular fitness, increases in muscle hypertrophy, and better physical functioning and QoL) [18]. As these benefits also contribute to the alleviation of fatigue, reduce the risk of cancer recurrence and mortality [19,20], and act as a protective factor against the development of other chronic diseases (e.g., diabetes, cardiovascular disease, obesity, and depression) [21], incorporating PA into cancer rehabilitation programmes is seen as an important treatment intervention for patients with cancer [22]. For instance, within women diagnosed with breast cancer [23], increased PA levels are positively associated with higher overall QoL scores and inversely associated with fatigue. Engaging in preoperative exercise programmes has also been shown to increase the physical fitness, QoL, and postoperative outcomes of cancer patients undergoing surgical interventions [24].

Despite the key role PA plays in the rehabilitation process, the purpose of cancer rehabilitation is multifaceted, and strong evidence surrounding the importance of incorporating psychosocial care into programmes has only recently been recognised [25]. Research on the biopsychosocial model of healthcare also supports this, as holistic multidisciplinary practices which provide social support, reduce emotional distress, and improve mental resilience are also shown to be protective factors for patients undergoing treatment [26]. As such, it is recommended that cancer rehabilitation services use the opportunity to implement psychosocial-oriented practices into their programmes to guide cancer patients towards better lifestyle choices and positive behaviour changes. In turn, higher survivorship of the disease, greater mental wellbeing levels, and improved QoL measures during and after treatment should be observed [27]. Furthermore, this form of care can also be used to encourage the uptake of, and adherence to programmes of exercise/PA; Mikkelsen, and colleagues [28] noted that social support and fixed conditions (e.g., group-based exercise classes, guided supervision, and coaching) were some of the strongest motivators and facilitators of PA for patients with cancer.

In addition to the above, mindfulness-based interventions (MBIs) have been shown to support positive changes in QoL, pain, fatigue, anxiety, and depression in cancer patients [29], and the cultivation of mindfulness, involving acceptance and nonjudgment of present-moment experience, often results in transformative health behaviour change (e.g., uptake and sustained PA behaviours). Evidence also suggests that MBIs are effective for reducing harmful health behaviours (i.e., prolonged sedentariness), thereby catalyzing chronic disease self-management and health behaviour change and improving physical and mental health outcomes [30].

Ensuring that effective oncological care is accessible and affordable will be crucial in the future development of successful cancer rehabilitation services [31]. However, as the structure, content, and delivery of these services vary between programmes, the effectiveness of individual interventions needs to be assessed. Therefore, the purpose of this evaluation was to assess the effectiveness of a recently established five-week (ten, 1 h sessions) group exercise, education, and mindfulness cancer rehabilitation programme (iCan), which focuses on supporting lifestyle and behaviour change and improving the fatigue indices, functional capacity, mental wellbeing, and QoL of participating patients with cancer.

## 2. Materials and Methods

Ahead of this service evaluation, retrospective ethical approval was obtained from the Faculty of Medicine and Life Sciences at the University of Chester (ref: 1888–22-CK-CMS).

### 2.1. iCan Cancer Rehabilitation Service

In development since 2018, a cancer rehabilitation service (iCan) designed and coordinated by Lifestyle Fitness & Physiotherapy and funded by the Lingen Davies Cancer Fund was launched in January 2022. The data presented in this service evaluation were routinely collected from people with cancer who attended iCan in the calendar year of 2022.

The iCan programme, which is linked directly into local clinical referral pathways, is based upon a multimodal approach that focuses on delivering holistic lifestyle support across online, face-to-face and hybrid sessions that are in accordance with patients’ needs. The iCan programme receives cancer patients through a single point of contact for the geographical areas covered from one of two pathways: (1) self-referral or (2) referral by specialist counsellors, primary care networks (general practitioners), social prescribers, cancer care navigators/coordinators and clinical teams in Shropshire, Telford and Wrekin, and Mid-Wales Health Trusts. Apart from a cancer diagnosis, there are no other specific eligibility criteria for enrolment in iCan. The programme is designed to be inclusive, thus maximizing accessibility to people living with cancer in the local community. It is noteworthy that the term *cancer survivor* is the common nomenclature for people living with cancer [32]; however, we use the terms *people living with cancer*, or *patient with cancer*, since some feel the term survivor has negative connotations relating to resilience and is perceived by many to be pessimistic in its outlook [33]. Patients can be referred at any point post cancer diagnosis and, through discussion with the iCan team, participants decide when they wish to begin. Participants are enrolled onto the programme to attend ten clearly defined cancer rehabilitation sessions over a five-week period, which comprise a group-based exercise/circuit programme, educational discussions surrounding a set theme or topic, and additional session-relevant mindfulness activities (Figure 1) aimed to support mental wellbeing and knowledge around managing cancer-related symptoms. This element of the programme is based upon an evidence-based behaviour change and motivational interviewing approach [34,35,36]. The structure of a typical session is presented in Appendix A.

Sessions may be accessed online and/or face-to-face in community settings, and each session includes a mindfulness-specific component. The sessions are led by an experienced team of exercise physiologists/scientists with advanced and cancer-specific exercise qualifications. The duration of the sessions is approximately one hour. Pre-session and follow-up support and resources are provided, and participants have direct access to staff support/contact. Whilst there was online provision available from iCan during the data collection period, we present only data from participants who attended in-person rehabilitation.

Each participant received one-to-one behaviour change support in which their goals and motivations were explored and barriers to participation identified, with appropriate planning implemented to negate them. Exercise programming and activity were adjusted to individual needs and co-morbidities, and each participant received support to modify exercise (within each session, and for any activities completed independently) as required. The final session (session 10) included a component where individual achievements were recognised and, as a follow-up, a further brief intervention around personal ownership and ‘what next’ goals were set. Each participant also received a review of outcome measures which were linked to progress versus their initial goals. Across the programme, social interaction was actively encouraged and supported throughout all sessions (face-to-face and online). Online sessions were available for individuals that preferred online contact, could not access sessions due to their rural location, or were unable to attend face-to-face sessions. Upon completion of the core programme, patients could access further extended/developed iCan rehabilitation programmes.

### 2.2. Data Collection

At the beginning and end of the five-week programme, functional capacity was assessed, and a series of previously validated questionnaires evaluating changes in mental health and wellbeing, domain-specific PA performance, and attitudes towards PA participation were administered to the participants for completion.

### 2.3. Outcome Measures

#### 2.3.1. Functional Capacity

The 30 s sit-to-stand test (STS_30_) is a basic test of physical functionality that provides a valid measure of functional capacity and lower body strength by counting the number of times an individual can squat into and out of a chair within 30 s [37]. The score of this test is taken as the ‘total number of squats completed within 30 s’ and has been shown to have high test–retest reliability (ICC = 0.948) in patients with cancer [38].

#### 2.3.2. Mental Wellbeing and Psychological Functioning

The Short Warwick–Edinburgh Mental Wellbeing Scale (SWEMWBS) is a 7-item scale of mental wellbeing and psychological functioning that focuses on positive elements of mental health. It has shown good content validity (Cronbach’s alpha = 0.91 in the general population) and high correlations with other mental health and wellbeing scales [39]. The 7 items ask participants to think over the last two weeks and respond using a 5-point Likert-based scale with a continuum from ‘none of the time’ to ‘all of the time’. Responses are summed to provide a total score (14–70), with higher scores representing better mental health and wellbeing.

#### 2.3.3. Physical Activity

The International Physical Activity Questionnaire (IPAQ)—Short Form was used to collect PA data from participants. The IPAQ was designed to measure health-related PA in population-based studies [40] and is widely used in clinical settings and PA research; it has previously shown good test–retest reliability (Spearman’s reliability coefficient *p* ~ 0.8) and reasonable criterion validity (pooled *p* = 0.30, 95% CIs of 0.26 to 0.39) when compared to accelerometery. The IPAQ asks participants to recall their last seven days of PA and equates this to four domains (vigorous-intensity, moderate-intensity activity, walking, and sitting) that are then converted into Metabolic Equivalent of Task minutes per week (MET-mins/wk) using formulae presented by Ainsworth et al. [41]. Accordingly, self-reported PA levels are presented in MET-mins/wk and/or actual minutes per day.

#### 2.3.4. Behaviour Change

The Capabilities, Opportunities, Motivation, Behaviour (COM-B) model [42] is cited by NICE as a key theoretical framework within which we can understand behaviour change [43]. To understand the determinants of PA within the current sample, a six-item questionnaire which measures these constructs was administered [44]. The questionnaire has shown acceptability, validity (floor effects 0.6–5.5% and ceiling effects 4.1–22.9%; pairwise correlations r significantly <1.0), and reliability (ICC: 0.554 to 0.833) for self-evaluation [44]. Each construct (COM) is represented by two questions, and each question asks the participant to respond to a question based on a 10-point scale (strongly disagree to strongly agree).

Participants were also asked to answer two questions surrounding self-perceived motivation and confidence levels using a 10-point Likert-based scale to rate (0 = very low; 10 = very high) ‘motivational importance’ and ‘current confidence levels’ at the beginning and end of the intervention. In addition to providing pre/post measures of change, the responses are actively used to support patients as they provide a platform upon which to explore individual behaviour change, as appropriate.

#### 2.3.5. Fatigue

The Functional Assessment of Chronic Illness Therapy (FACIT) Fatigue Scale is a 13-item tool that measures an individual’s level of fatigue during their usual daily activities over the past week [45]. The level of fatigue is measured on a four-point Likert scale (4 = not at all fatigued to 0 = very much fatigued). The FACIT Fatigue Scale has been widely utilized by clinicians and researchers [46] and reportedly has high internal validity (Cronbach’s alpha = 0.96) and high test–retest reliability (ICC = 0.95) in those with chronic disease [47].

#### 2.3.6. Data Analysis

Available case analysis was used to identify and remove any individual datasets that were missing pre- and/or post-timepoint measures. Assumptions of normality were checked using the Shapiro–Wilk test, and descriptive statistics (median and interquartile range) were calculated for each variable; a Wilcoxon rank test was used to identify statistical differences (*p* < 0.05) between the pre- and post-intervention time points for each outcome measure, and effect sizes (Cohen’s d) were reported. All statistical analysis was completed using Jamovi data analysis software (The Jamovi Project 2023, Jamovi, Version 2.3).

## 3. Results

In this service evaluation, the datasets of 128 iCan participants (males, *n* = 34; females, *n* = 94) were retrospectively analyzed. The mean age of the participants included in this evaluation was 65 ± 9.7 years, and most of the participants were post-treatment (Table 1) and had a diagnosis of breast or prostate cancer (Table 2). To be included in this evaluation, patients were required to complete 80% (8/10) of the rehabilitation sessions over a 5-week period. It is pertinent to highlight that for participants in the initial iCan cohorts (including participants whose data are reported in this service evaluation), the mean number of sessions attended was 8.3 (83%). In addition to a high level of attendance, it is also noteworthy that there have been zero adverse events reported from any participant/cohort who have attended iCan via any modality.

### 3.1. Physical Outcome Measures

#### Functional Capacity, Physical Activity, and Fatigue

Following the iCan intervention, a statistical improvement occurred between the pre and post STS_30_ scores (*p* < 0.01), with participants (*n* = 59) performing an average of 6.3 additional repetitions over the 30 s period (Table 3). Statistical differences were also observed across the IPAQ (Table 3), with iCan participants engaging in a greater number of moderate and vigorous activities, increasing the amount of time being physically active, and reporting lower levels of sedentary behaviour. These increases in PA also meant that 49 participants improved their PA classification [i.e., they moved from inactive to minimally active, inactive to performing health-enhancing PA (HEPA), or from minimally active to HEPA); some 44 maintained their PA classification, and only 2 were classified in a lower category (Appendix A). Similarly, when FACIT scores were analyzed, a statistical decrease in fatigue (*p* < 0.001) was observed (Table 3).

### 3.2. Psychological Outcome Measures

#### Mental Wellbeing, Psychological Functioning, and Behaviour Change

The results of the SWEMWBS showed that the mental wellbeing of iCan participants (*n* = 91) improved over the 5-week intervention period (Table 3). Similarly, self-perceived importance levels (*n* = 80) and self-reported confidence levels (*n* = 81) also saw statistically significant improvements (Table 3). Statistical increases from baseline values were observed for all COM-B domain scores (Table 3) after the 5-week intervention had been completed.

## 4. Discussion

Service evaluations can be fraught with challenges such as financial implications, use of resources, and challenges in implementing recommendations (e.g., the practicality of recommendations, disruption to services, and staff training needs) [40]. By contrast, service evaluations in healthcare settings also serve as an important mechanism for improving the standard of patient care (e.g., enhancing patient safety or identifying where improvements are needed), evaluating effectiveness, optimizing utilization of available resources, ensuring regulatory compliance, and identifying the training needs of staff [48]. This service evaluation sought to review the effectiveness of the five-week iCan cancer rehabilitation ‘core’ programme. We conducted a retrospective analysis which aimed to evaluate the impact of iCan on improving the fatigue indices, functional capacity, and mental wellbeing of people living with cancer. In addition, it sought to better understand whether participants’ readiness to engage in PA and their PA behaviours were improved by the iCan programme. After analysis, it was identified that the iCan cancer rehabilitation programme was an effective method of improving the wellbeing of people living with cancer, with notable improvements occurring across each of the physical and psychological measures included in this evaluation.

The post values reported for the STS_30_ suggest that participation in five weeks of iCan had a beneficial effect upon functional capacity. When the scores from baseline were considered, it is apparent that, compared to the general population, the mean scores were below normative scores regardless of both age and gender, suggesting impaired functional capacity prior to participation [49]. Whilst that is perhaps unsurprising, it is encouraging that following the iCan programme, mean scores were increased to levels that are classed as being in a normal range. However, understanding the importance of this improvement is difficult without having a universally established value for the STS_30_’s minimum clinically important difference (MCID).

Given multiple factors (i.e., cancer-related symptoms, treatment stage, and baseline functional status), it is difficult to determine the MCID in patients with cancer. Previous studies reporting on other populations have reported STS_30_ MCID values ranging from ≥2 repetitions in subjects with moderate-to-severe chronic obstructive pulmonary disease [50], up to three repetitions (IQR: 1, 6) in older adults’ change in score between leaving an intensive care unit and hospital discharge [51]. Whilst these data are from populations with severe functional impairment, the magnitude of change following five weeks of iCan is certainly encouraging and likely to indicate that functional capacity has improved.

The other measure of functional capacity captured by iCan is the PA level/classification of participants as measured by self-report using the IPAQ. The beneficial effects of PA for cancer survivors’ health are widely reported [52], and encouragingly, there were statistically favourable effects across each PA domain, with more days spent engaging in PA, more minutes of PA per day (except walking), and more MET minutes per week being performed; there was also a statistical reduction in daily sitting time. Before starting the iCan programme, ~38% of participants were classed as inactive, and only ~17% completed enough PA for it to be considered HEPA (Appendix A). It is reported that PA levels may fall after cancer diagnosis. For example, breast cancer patients have been noted to be active for two hours less per week compared to pre-diagnosis. This is further exacerbated for patients receiving chemo/radiotherapy, in whom pre-diagnosis PA levels may fall by 50% [53]. After five weeks in the programme, the number who were inactive reduced to ~7%, and those performing HEPA increased to ~44%, indicating a general trend for participants doing more PA each week; in fact, only two participants were classified as performing less weekly PA after attending iCan, which is likely related to either their treatment or a worsening of symptoms.

Whilst ~48% of participants were still classified as minimally active following iCan (this is 32.3% in the general UK population), it is important to acknowledge that performing some PA is better than doing none [54], and iCan patients are directly receiving a minimum of two individualized sessions per week. It is of interest to note that maintaining and/or increasing PA levels as seen in iCan measures should be considered significant alongside the reported falls in activity that are associated with diagnosis and treatments. In fact, large meta-analyses of studies from the USA National Cancer Institute reported that performing some MVPA, but less than recommended levels, was associated with a 20% reduction in all-cause mortality risk [55] and an increase of 1.8–2.5 years in life expectancy [56] when compared to those performing no leisure-time MVPA.

Other important guidelines regarding PA levels for patients with cancer are that as few as two episodes of activity that may support strength and balance benefits should be achieved each week. It is noted that only 46% of people with a long-term health condition (including cancers) are meeting strength health guidelines, falling to just 36% of those with three or more co-morbidities [57]. Whilst these elements are not separately measured as part of iCan outcomes, all iCan participants are achieving this guideline by taking part in the twice-weekly sessions, as strength- and balance-based exercises are included as core components.

The self-reported increases in PA and the improved PA classification of participants may be explained, at least in part, by the increases to participants’ perceived capability, opportunity, and motivation to be physically active; these aspects are strongly reinforced and addressed throughout the iCan programme. Michie and colleagues [32] suggest that it is (physical and psychological) capability, (physical and social) opportunity, and (automatic and reflective) motivation that constitute the essential conditions for uptake and maintenance of a given behaviour. When that behaviour is specifically PA, previous studies have tried to quantify which of the constructs has the biggest effect upon PA behaviours. A prospective study in healthy adults identified that higher capability and motivation were the key drivers of PA behaviour [18], whereas other studies also identified reflective motivation but with either physical opportunity [58] or automatic motivation [59] as the strongest predictors of PA behaviour.

There are a number of reviews which have reported on these constructs across a range of cancer types; physical capability and opportunity, alongside automatic motivation, were the components identified in patients with head and neck cancer [60], whilst physical capability was also identified in patients with lung cancer [61]. It is likely that variations in personality, personal circumstances, and the nature of cancer symptoms and treatment side effects will impact upon an individual’s capability, opportunity, and motivation to engage in PA, and even that individual levels may fluctuate on a day-to-day basis. However, the multicomponent nature of iCan (i.e., supervised exercise, PA and information/education sessions, resources and direct support, mindfulness, and health assessments) is likely to have a beneficial effect upon each of the constructs and of course, the volume of PA participants engage in. Whilst a universally accepted measure of COM-B is lacking, and there is no definition of what constitutes an important change in the scale created by Keyworth and colleagues [34], the increase in all six scores and corresponding increase in PA levels following five weeks of iCan do seem to be promising, supporting the delivery methods and overall design of the iCan programme.

This service evaluation also revealed statistically reduced sitting time upon completion of the iCan programme. This is particularly important since an individual’s risk of premature mortality may also be dependent upon the amount of sedentary behaviour performed. A recent study in cancer survivors [62] found that increased sitting time was associated with higher risks in all-cause and cancer-specific mortality, and that those who were at the greatest risk were those who were sitting for the longest times whilst also not engaging in sufficient levels of PA (all-cause mortality hazard ratio: 5.38; 95% CI, 2.99–9.67). It has also been reported that those who replaced 30 daily minutes of sedentary behaviour with either light-intensity (e.g., walking) or MVPA had relative risk reductions in all-cause mortality of 20% and 51%, respectively [63]. This is particularly encouraging for participants who have completed iCan because average daily sitting time is reduced by ~100 min, whilst there are concomitant increases in MVPA which are likely to be equivalent to more than 30 daily minutes.

Upon completion of iCan, there was also a statistically favourable change in average FACIT-Fatigue scores, indicating that participants felt less fatigued. When the post-iCan scores were compared to scores from normative population scores (Germany: 43.5 ± 8.3; USA: 43.6 ± 9.4 and 46.6 ± 7.2), they were unsurprisingly still markedly lower [64,65,66]. What is apparent however, is that scores prior to iCan were also markedly lower than previously reported FACIT scores from patients with cancer, and it was not until post-iCan FACIT-Fatigue scores were considered that they become comparable to other studies. Previous studies have reported FACIT-Fatigue scores in patients with cancer to be 36.9 [66], 34.6 [67], and 40.0 [65], but these are not reported in response to any rehabilitation/exercise regime. It has, however, been reported that aerobic and resistance exercise have a beneficial effect on cancer-related fatigue [68,69] and that a three-point change has previously been identified as a clinically important difference [70]. Given the magnitude of change observed following iCan (i.e., ~7.5 points), it is reasonable to assume that the programme has had an important beneficial effect on cancer-related fatigue and that subsequently this may also be contributing to the increase in health-protective behaviours (i.e., increased PA and reduced sedentary behaviour) as reported above.

There was also an observed beneficial effect upon mental wellbeing after five weeks of iCan. Whilst the magnitude of the change in SWEMWBS score appears to be relatively small (7.8%), it is noteworthy that the participants, even pre-iCan, had higher scores than those reported from data representative of the English general population [71]. The range of scores around the mean (4.18–4.25) does indicate that some patients started iCan with indications of low mental wellbeing and that there were positive changes over the five weeks that moved the range to within the population’s average range. What is perhaps more telling of the benefit is that the MCID for SWEMWBS is thought to lie between 1 and 3 points difference [72], within which the average improvement (+2.07 points) from pre- to post-iCan firmly falls, indicating significant value for patients.

It should also be noted that part of the iCan assessment and patient pre-engagement process includes the use of behaviour change techniques based upon motivational interviewing. During initial communications, a brief intervention (behaviour change) format was followed, and this may have affected pre-programme measures, given that brief intervention techniques may have had a positive impact (e.g., awareness, motivation, and confidence). The purpose of early contact (normally telephone based) was to provide information about the iCan programme, raise awareness of benefits, dispel fears/myths (e.g., regarding risks), and to support confidence and motivation for behaviour change.

As a result of this brief intervention (delivered to all referred patients), pre-iCan outcome measures may have differed from responses obtained prior to any contact (i.e., there may have already been a positive impact that will be detected in pre-measures baselines). This initial contact approach was felt to be a critical part of establishing patient support on their terms and meeting their specific needs.

### 4.1. Limitations

Whilst there were several notable benefits to participants’ functional capacity and mental wellbeing after participating in the iCan programme, there are some limitations to our evidence, which should be discussed. The first of these is the study design; in fact, the iCan programme was never intended to be a research study. Moreover, this programme of rehabilitation was designed using an evidence-based approach and years of experience contained within the delivery team. This service evaluation reports on outcomes which were collected as part of the routine delivery of the programme so as to be able to monitor effectiveness. Whilst there appears to be a beneficial effect following iCan, it is not possible to demonstrate the cause–effect relationship between the intervention and outcomes. Although no study is truly able to prove causality, randomization to iCan or to a control group would be the gold-standard design for evaluating such an intervention [73].

Except for the STS_30_, all other reported data are self-reported, which may present a methodological concern. For example, social desirability bias and/or recall bias may mean that there is an increased risk that participants over-report their PA levels [74] and under-report their sedentary time [75]; such an effect has been previously reported in Spanish cancer survivors when the IPAQ-SF was compared to device-measured PA and sedentary behaviour [76]. However, a methodology which incorporates self-reported outcomes, often alongside performance-based outcomes, is not only commonplace in health research but also within health and rehabilitation initiatives which aim to improve/preserve the health of participants [77]. The use of self-report measures is preferred because they are practical, low-cost, place a minimal burden on the participant, are generally accepted by participants, and can measure a range of constructs [78]. Further work could be done to explore the validity and reliability of such measures in cancer survivors, but that is beyond the remit of this evaluation and the iCan programme.

It is also notable that whilst the outcome measures captured by the iCan programme are deemed to represent functional capacity (i.e., STS_30_ and IPAQ) and QoL (i.e., SWEMWBS and FACIT), they are not absolute outcome measures for these constructs. For example, additional measures of physical function could have been taken or cancer-specific measures of QoL utilized in these participants. Whilst additional measures which better reflect these constructs could be added, it is important that they are implemented in such a way that no additional burden is placed upon those participating in the programme. The retrospective nature of the current evaluation meant that it was not possible to amend outcome measures, but for future studies/ongoing provision, scheme coordinators should work alongside referring healthcare professionals and patient groups to determine the most relevant/important outcomes.

It is notable that there are a significant number of datasets missing for participants that completed iCan. Considering that this evaluation includes 128 participants, one outcome (STS_30_) has complete data for only 59 participants (54% missing data). For other outcomes, missing data range from 37.5% to 25.8% which may introduce attrition bias. It is not clear whether there are systematic differences in those who completed various outcome data pre- and post-iCan, and given the large number of data missing from the performance-based outcomes, this could be important. It may, however, indicate that there are procedural elements in the iCan process that need to be reviewed to ensure that the maximum amount of data can be collected for all participants. This evaluation may provide a step upon which to base research programmes around this clear delivery intervention.

### 4.2. Future Recommendations

Due to the timing and method of data collection within patient pathways, it has been noted that a system that is simplified for the patient may be useful. The early stages of diagnosis and treatment that many patients are in at the point of referral to iCan make many demands upon their time, and it is recognized that completing measures may not be overly motivating. Therefore, online processes are being developed for iCan data collection to ensure greater compliance and full completion of pre- and post-outcome measures.

As iCan is demonstrated to be successfully providing benefits for cancer patients, a future recommendation may be to incorporate ‘return-on-investment’ types of analysis. This is likely to be relatively complex, given the scope and extent of wider iCan delivery, but would provide comparative information on this community-delivered programme. Retrospective patient audit type approaches may be a starting point from which to establish post-programme care access needs and savings on treatment cost.

There are many facets to the delivery of iCan that may offer compounded benefits; there are also a wide range of patients that may access the process. It is recommended that future research explores the relative effectiveness of iCan across these facets (e.g., PA, education, behaviour change, and mindfulness) and with different patient groups (e.g., cancer type, stage, treatment status, gender, age, etc.). Regarding the latter, it would be beneficial to include individual analyses, perhaps as part of a sufficiently powered randomized controlled trial, depending upon a patient’s diagnosis and their place on the cancer care continuum (i.e., pre, during, or post treatment).

The ‘core’ iCan programme referred to in this paper is an important part of a much wider iCan delivery process that is designed to offer high-quality information, support, advice, and personalized care specific to participants’ cancer and their needs in order to enhance services and improve lives. Future research and evaluation of the additional components and their interaction with the core iCan programme may offer useful insight to the longer-term lifestyle outcomes experienced by cancer patients in Shropshire, Telford and Wrekin, and Mid-Wales.

## 5. Conclusions

The core iCan programme is a clearly structured and defined five-week multicomponent cancer rehabilitation programme, which is part of a much wider iCan delivery process that supports cancer patients for 12 months. The core iCan programme operates in a socially interactive community-based environment and incorporates supervised exercise, PA education, mindfulness, and behaviour change with the aim of improving the health and wellbeing of cancer survivors. The highly qualified and experienced staff that deliver the programme ensure that each cohort delivery maintains programme fidelity, as defined in the programme design parameters. Having a clearly defined and consistent delivery process is understood to be uncommon and enhances confidence in the future reproducibility of core iCan. From the data collected in the programme, and consequently the outcomes reported in this evaluation, it appears that iCan is indeed having beneficial effects upon the physical/functional and psychological health of its participants. Where data are available, there appear to be clinically significant improvements across the range of measured, functional, wellbeing, activity/sedentariness outcomes, which suggest that participation in iCan is instrumental in adding value to the health and wellbeing of the patients of Shropshire, Telford and Wrekin, and Mid-Wales.

By encouraging, supporting, and empowering individual cancer patients to adopt healthier lifestyles and reduce risks, iCan is contributing to improving short/longer-term outcomes and reducing the burden of a patient’s cancer diagnosis, treatment, and recovery at all points on the pathway. iCan provides the framework, process, and availability to provide adaptable and effective cancer rehabilitation support for patients of Shropshire, Telford and Wrekin, and Mid-Wales.

## Figures and Tables

**Figure 1 diseases-12-00236-f001:**
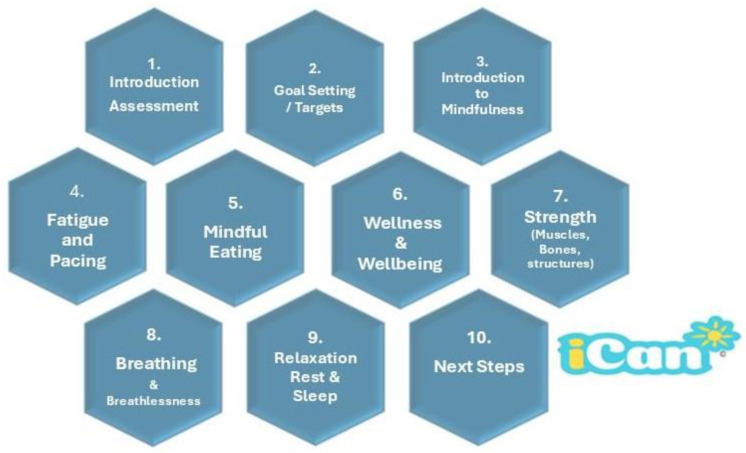
Structure of iCan core sessions.

**Table 1 diseases-12-00236-t001:** Mean (±SD) age of participants and frequency (%) of cancer treatment stage within the iCan cohort (*n* = 128).

Outcome	Total	Female	Male
Age (years)	65.0 ± 9.7	62.6 ± 9.2	71.4 ± 7.8
*Treatment stage* (*n*)			
Pre	3 (2.3%)	3 (3.2%)	-
During	54 (42.2%)	35 (37.2%)	19 (55.9%)
Post	71 (55.5%)	56 (59.6%)	15 (44.1%)

**Table 2 diseases-12-00236-t002:** Frequency of cancer types within the iCan cohort (*n* = 128).

Cancer Type	Frequency	Percentage
Breast	76	59.4%
Prostate	22	17.2%
Head and Neck	6	4.7%
Bowel and Colorectal	6	4.6%
Other	18	14.1%

**Table 3 diseases-12-00236-t003:** Comparison of reported outcomes pre- and post-iCan.

Outcome	Pairs (*n*)	Pre iCan	Post iCan	Cohen’s *d*	*p* Value
30 s Sit-to-Stand	59	8.0 (6.0–9.0)	14.0 (11.5–17.0)	−1.00	<0.001
FACIT	93	29.3 (21.0–39.0)	37.0 (28.0–42.0)	−0.77	<0.001
SWEMWBS	91	25.0 (22.5–27.0)	26.0 (23.0–29.0)	−0.69	<0.001
Importance	80	9.0 (8.0–10.0)	10.0 (8.75–10.0)	−0.44	0.016
Confidence	81	8.0 (6.0–9.0)	9.0 (8.0–10.0)	−0.76	<0.001
*Physical Activity (days/wk)*					
Vigorous Activity	95	0.0 (0.0–0.0)	0.0 (0.0–2.0)	−0.78	<0.001
Moderate Activity	95	0.0 (0.0–3.0)	3.0 (2.0–5.0)	−0.81	<0.001
Walking	95	4.0 (2.0–7.0)	5.0 (3.5–7.0)	−0.57	<0.001
*Physical Activity (mins/day)*					
Vigorous Activity	95	0.0 (0.0–0.0)	0.0 (0.0–45.0)	−0.73	<0.001
Moderate Activity	95	0.0 (0.0–60.0)	45.0 (30.0–82.5)	−0.65	<0.001
Walking	95	60.0 (30.0–60.0)	60.0 (30.0–60.0)	−0.10	0.506
Sitting	95	360.0 (300.0–480.0)	300.0 (240.0–360.0)	0.85	<0.001
*Physical Activity (MET-mins/wk)*					
Total Activity	95	933.0 (479.0–2318.0)	2106.0 (1263.0–3593.0)	−0.76	<0.001
Vigorous Activity	95	0.0 (0.0–0.0)	0.0 (0.0–720.0)	−0.81	<0.001
Moderate Activity	95	0.0 (0.0–680.0)	720.0 (360.0–1350.0)	−0.69	<0.001
Walking	95	693.0 (256.0–1172.0)	693.0 (363.0–1386.0)	−0.37	0.008
*COM-B domain scores*					
Physical Opportunity	89	8.0 (6.0–9.0)	8.0 (7.0–10.0)	−0.46	0.002
Social Opportunity	89	7.0 (6.0–9.0)	8.0 (6.0–9.0)	−0.32	0.028
Reflective Motivation	89	7.0 (5.0–8.0)	8.0 (6.0–9.0)	−0.56	<0.001
Automatic Motivation	89	5.0 (4.0–7.0)	6.0 (5.0–8.0)	−0.56	<0.001
Physical Capability	89	6.0 (5.0–8.0)	8.0 (7.0–9.0)	−0.73	<0.001
Psychological Capability	89	7.0 (6.0–8.0)	8.0 (7.0–9.0)	−0.59	<0.001

Key: FACIT: Functional Assessment of Chronic Illness Therapy; SWEMWBS: Short Warwick–Edinburgh Mental Wellbeing Scale; MET-mins/wk: metabolic equivalent of task—minutes per week; COM-B: Capability, Opportunity, Motivation and Behaviour; all data are reported as median and interquartile range; a paired-samples *t*-test using the Wilcoxon W non-parametric method was performed on all outcomes; statistical significance set at *p* < 0.05.

## Data Availability

The datasets presented in this article are not readily available because the data were obtained by the authors specifically to conduct this service evaluation as permitted by the ethical approval granted by the University of Chester. Requests to access the datasets should be directed to the corresponding author.

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
