# Peer review of "iCan, Empowering Recovery: Evaluating a Patient-Centred Cancer Rehabilitation Programme across the Cancer Care Continuum"

_diseases, 2024, doi:10.3390/diseases12100236_

Round 1

Reviewer 1 Report

Comments and Suggestions for Authors

Cancer rehabilitation is an important part of treatment. The author of this article focuses on cancer rehabilitation and conducts systematic research using the established iCan rehabilitation service, revealing new findings in rehabilitation service evaluation. The quality and innovation of the paper are both good, and the experimental results are reliable. I have a few minor revision suggestions here, and the review comments are as follows:

1. The image quality of the paper needs to be improved. Please provide paper images with a resolution of 300 dpi or higher.

2. I suggest the author add a discussion and analysis of the advantages and disadvantages of this research method in the Discussion section.

Author Response

Reviewer 1: Comments to the Author

Cancer rehabilitation is an important part of treatment. The author of this article focuses on cancer rehabilitation and conducts systematic research using the established iCan rehabilitation service, revealing new findings in rehabilitation service evaluation. The quality and innovation of the paper are both good, and the experimental results are reliable. I have a few minor revision suggestions here, and the review comments are as follows:

  1. The image quality of the paper needs to be improved. Please provide paper images with a resolution of 300 dpi or higher.

  1. I suggest the author add a discussion and analysis of the advantages and disadvantages of this research method in the Discussion section.

Reply:  We thank the reviewer for the time dedicated to reviewing our paper and for the positive feedback and helpful comments/suggestions. Accordingly, we have made the following revisions based on the provided comments:

  1. We have provided a higher quality image (300 dpi) of the image in Figure 1.
  2. We have added in some text which highlights the role of service evaluations in the context of healthcare. This includes some of the challenges faced (disadvantages) and benefits of them being conducted. The new text is at the start of the discussion (lines 315-321).

Reviewer 2 Report

Comments and Suggestions for Authors

Dear authors your work and effort in cancer rehabilitation field is undoubtedly of extreme importance both for the patients and cancer rehabilitation professional. Unfortunately in your paper you haven't succeeded to prove your thesis and value of your iCan programme due to substantial methodological an statistical flows. As I am aware that your programme didn't start as research study, let alone RCT, still evaluation of the program should be and can be done more profoundly.

In order to help you in your future professional and scientific work I suggest the following:

1. In methodology section:

Firstly selection of participants (exclusion and inclusion criteria etc) should be described in details. Participants demographic details as well as disease, treatment and impairments characteristics should be stated in details.

Participants are not well selected and divided. You have to few participants in pre - treatment group and I strongly advise you to exclude them from the study because you cant make any sound conclusion based on only 6 participants. Also you have too few participants with carcinomas other than breast and prostate cancer so you should consider what to do with them based on impairments they have. As fare as I can conclude from scarce data you have provided the best way to divide your participants is either by status of cancer treatment (treatment and post treatment group) or by impairments they have before entering your programme. This is the only way in which you could evaluate is your programme suitable for all impairments and throughout cancer treatment continuum. 

Secondly more details about programme composition and providers should be given. Please specify how the program is patient tailored. Problem is that programme can be online, on site - face to face and in small groups because from the published literature we already now that there is difference in adherence, quality as well as in the successfulness of these different rehabilitation approaches. Please specify how are they led and by whom. How many participants attended on line, face to face or combined programme?..

Programme goals are not precisely stated. Each goal should be connected to proper outcome measure. Please chose specified goals that you have proper outcome measure. For example you don´t have measure for cancer patients QOL. Also for functional capacity you should add if possible Eastern Cooperative Oncology Group (ECOG) performance Scale. 

We don´t know the period when the program started for specified participants and for how long data were collected. 

Provide references upon each part of the programme was based on. For example exercise - Turner RR, Steed L, Quirk H, Greasley RU, Saxton JM, Taylor SJ, Rosario DJ, Thaha MA, Bourke L. Interventions for promoting habitual exercise in people living with and beyond cancer. Cochrane Database Syst Rev. 2018 Sep 19;9(9):CD010192. doi: 10.1002/14651858.CD010192.pub3. PMID: 30229557; PMCID: PMC6513653.

Watson G, Coyne Z, Houlihan E, Leonard G. Exercise oncology: an emerging discipline in the cancer care continuum. Postgrad Med. 2022 Jan;134(1):26-36. doi: 10.1080/00325481.2021.2009683. Epub 2021 Dec 20. PMID: 34854802.

Throughout the paper you are constantly inadequately mentioning different concepts that are of you interest like functional capacity, patient well being, quality of life etc.

You should revise Results and discussion sections accordingly.

2. In Introduction section cancer care continuum should be described.

Mayer RS, Engle J. Rehabilitation of Individuals With Cancer. Ann Rehabil Med. 2022 Apr;46(2):60-70. doi: 10.5535/arm.22036. Epub 2022 Apr 30. PMID: 35508925; PMCID: PMC9081390.

Rezende, G., Gomes-Ferraz, C. A., Bacon, I. G. F. I., & De Carlo, M. M. R. do P. (2022). The importance of a continuum of rehabilitation from diagnosis of advanced cancer to palliative care. Disability and Rehabilitation45(24), 3978–3988. https://doi.org/10.1080/09638288.2022.2140456

https://ascopubs.org/doi/10.1200/EDBK_349635

Distinction between cancer patients and cancer survivors should be made.

Adverse effects and impairments following cancer treatment should be described in more details according to your programme goal.

Also you should provide some references about similar cancer rehabilitation programmes and state the differences between yours and existing as wall as what you believe is innovative about your program. For example Lopez CJ, Santa Mina D, Tan V, Maganti M, Pritlove C, Bernstein LJ, Langelier DM, Chang E, Jones JM. CaRE@ELLICSR: Effects of a clinically integrated, group-based, multidimensional cancer rehabilitation program. Cancer Med. 2024 Feb;13(4):e7009. doi: 10.1002/cam4.7009. PMID: 38457258; PMCID: PMC10923049.

Also explain benefits of telemedicine in cancer rehabilitation Davidoff C, Cheville A. Telemedicine in Cancer Rehabilitation: Applications and Opportunities Across the Cancer Care Continuum. Am J Phys Med Rehabil. 2024 Mar 1;103(3S Suppl 1):S52-S57. doi: 10.1097/PHM.0000000000002421. Epub 2023 Dec 21. PMID: 38364031.

Title should be changed so that is obvious that rehabilitation is provided throughout cancer care continuum.

Comments on the Quality of English Language

Please do perform detailed grammar and sentence construction check up.
